# An actor-model framework for visual sensory encoding

Franklin Leong ®[1], Babak Rahmani[2,5], Demetri Psaltis ®[3], Christophe Moser ®[2] & Diego Ghezzi ®[1,4] ✉

A fundamental challenge in neuroengineering is determining a proper artificial input to a sensory system that yields the desired perception. In neuroprosthetics, this process is known as artificial sensory encoding, and it holds a crucial role in prosthetic devices restoring sensory perception in individuals with disabilities. For example, in visual prostheses, one key aspect of artificial image encoding is to downsample images captured by a camera to a size matching the number of inputs and resolution of the prosthesis. Here, we show that downsampling an image using the inherent computation of the retinal network yields better performance compared to learning-free downsampling methods. We have validated a learning-based approach (actor-model framework) that exploits the signal transformation from photoreceptors to retinal ganglion cells measured in explanted mouse retinas. The actor-model framework generates downsampled images eliciting a neuronal response in-silico and ex-vivo with higher neuronal reliability than the one produced by a learning-free approach. During the learning process, the actor network learns to optimize contrast and the kernel's weights. This methodological approach might guide future artificial image encoding strategies for visual prostheses. Ultimately, this framework could be applicable for encoding strategies in other sensory prostheses such as cochlear or limb.

Sensory organs capture information from the environment and convert it into neuronal signals that are interpreted by the brain during cognition. This transformation is known as sensory encoding.

Similarly, a sensory prosthesis converts information from sensors into artificial stimulation parameters to replace natural functions (artificial sensory encoding). However, prosthetic devices typically have an input range much smaller than biological systems. For example, the number of electrodes in neural implants is usually several orders of magnitude lesser than the number of sensory neurons[1,2]. Therefore, artificial sensory encoding is a form of dimensionality reduction. High-dimensional information from sensors is reduced to a few stimulation parameters while trying to maximally preserve the information, so that few electrodes can write information in a format that the brain can read and understand. A notable example is auditory encoding in cochlear implants, where sound is converted into electrical stimulation of a few frequency regions within the auditory nerve[3–5]. This process allows deaf individuals to hear sound. Likewise, limb prostheses provide amputees with tactile feedback to enhance manual dexterity, increase prosthesis embodiment, and improve their quality of life[6–8].

[1]Medtronic Chair in Neuroengineering, Center for Neuroprosthetics and Institute of Bioengineering, School of Engineering, École Polytechnique Fédérale de Lausanne, Geneva, Switzerland. [2]Laboratory of Applied Photonics Devices, Institute of Electrical and Micro Engineering, School of Engineering, École Polytechnique Fédérale de Lausanne, Lausanne, Switzerland. [3]Optics Laboratory, Institute of Electrical and Micro Engineering, School of Engineering, École Polytechnique Fédérale de Lausanne, Lausanne, Switzerland. [4]Ophthalmic and Neural Technologies Laboratory, Department of Ophthalmology, University of Lausanne, Hôpital ophtalmique Jules-Gonin, Fondation Asile des Aveugles, Lausanne, Switzerland. [5]Present address: Microsoft Research, Cambridge, UK. ✉e-mail: info@ghezzilab.org

Artificial sensory encoding plays a huge role also in visual prostheses. Artificial visual encoding converts high resolution images captured by an external camera into a spatiotemporal pattern of artificial stimuli delivered by a retinal[9–12], optic nerve[13,14], or cortical prosthesis[15,16]. Artificial visual encoding is critical to improve the patient's perception, but it is not straightforward. In the retina, information flows from approximately 120 million photoreceptors to roughly 1.2 million retinal ganglion cells (RGCs) divided into several classes, which project to several brain nuclei, including the lateral geniculate nucleus and then to the visual areas where further image processing occurs[17–20]. The complexity of the visual information process requires advanced encoding strategies to ensure effective stimulation of the visual neurons leading to a useful artificial vision.

To date, there have been several visual prostheses implanted in patients[16,21–26], but most devices were tested to recognize only letters and shapes using simple image encoding techniques (e.g. pixel averaging). In the Argus® II, the most implanted device so far, pixel averaging was used in conjunction with video filters to downsample the camera image to the resolution of the implanted array (6 × 10 pixels). Recently, there has been considerable research dedicated to the development of better image encoding algorithms. Some approaches include object detection, edge detection, and content-aware retargeting method[27,28]. In general, such methods aim to reduce the complexity of the image and highlight interesting content and features. For example, edge detection may identify the discontinuity of brightness in an image to locate the outline of an object. By reducing the amount of information, the user could better perceive the environment. However, these algorithms do not consider the retinal information processing from photoreceptors to RGCs. Therefore, their encoding potential might be limited.

Finding the proper artificial input to a sensory system that elicits the desired perception is an ill-posed problem: there are multiple inputs that could possibly yield the same output. In a linear system, the ability to produce a desired output can be determined by measuring the system's response to a series of arbitrary inputs (forward pathway) and then computing its inverse (backward pathway). Determining the forward pathway would entail measuring the responses of the whole system, which is practically impossible in a biological system given the large number of cells and the low number and resolution of measuring electrodes. Yet, even if the system's forward pathway is fully characterized, for large scale systems involving many variables, obtaining its inverse (backward pathway) is computationally intensive. Moreover, neuronal processing is non-linear, and it is probed only with partial measurements, thus further complicating this problem.

There had been significant efforts to generate in-silico retina models (forward pathway) that potentially could be used for efficient artificial visual encoding[29–32]. Retinal information processing is complex[33,34], and finding a high-performing in-silico retinal model is critical since it will directly impact the outcome of the image encoding algorithm. In recent years, convolutional neural networks (CNNs) have been very successful at modeling the retina and outperformed conventional approaches such as linear-nonlinear models or generalized linear models[35,36], which are less effective in capturing the retinal dynamics when white noise and natural scenes are presented[35,36]. Hence, using CNNs to model the retina presents a great potential in improving artificial visual encoding.

Still, a computational method to estimate the backward pathway, given the limited set of measures obtained in the forward pathway, is necessary for artificial visual encoding. Again, neural networks perform well in solving ill-posed inverse problems[37]. Therefore, we propose an end-to-end neural network-based approach for both retina modeling and image encoding which considers the retinal information processing. We validate an actor-model framework designed to learn non-linear downsampling patterns through a learning-based approach[37,38]. Performance is assessed through the measurement of

neuronal reliability[35]. By integrating the measured retinal information processing into the framework, we demonstrated, in-silico and ex-vivo, that the generated downsampled images elicit a neuronal response with higher neuronal reliability (+4.9% in-silico and +2.9% ex-vivo, median percent increase) compared to a learning-free approach (i.e. pixel averaging). During the process, the actor network learned to optimize contrast and the kernel's weights evolved towards a Mexican hat shape which resembles the receptive field (RF) of RGCs. These properties work in conjunction to enable effective downsampling.

The actor-model framework used in this study is general and could be exploited for other image encoding processes or even in other fields of artificial sensory encoding, such as auditory and tactile. Albeit belonging to different sensory pathways, auditory and tactile sensations share similar properties which allow us to postulate the potential effectiveness in other sensory encoding systems[34,39]. This learning-based approach could serve as a template for future encoding strategies accounting for the natural transformation process occurring in the sensory organ. A more effective encoding method entails that the brain could better interpret the encoded information, leading to improved perception of a prosthesis user.

## Results
### The actor-model framework in retinal processing

The actor-model framework is built following a 3-step approach (Fig. 1). In step 1, we projected a set of high-resolution images $\mathbf{X} \in \mathbb{R}^{128x128}$ (128 × 128 pixels) to mouse retinas explanted over a transparent multielectrode array (MEA) used to detect neural spikes from RGCs in response to image projection. For each identified RGC, we built a response vector by summing up the average number of spikes in response to image projection (Fig. 1, inset). Then, we combined the response vectors from each recorded RGC across multiple retinas into a neural response matrix forming the ground truth response $\mathbf{Y} \in \mathbb{R}^{nxm}$ where $n$ is the number of RGCs ($n = 60$ RGCs from $N = 10$ retinas) and $m$ is the number of images projected ($m = 1200$). We refer to the explanted retinas as the biological system ($\mathscr{F} : \mathbf{X} \rightarrow \mathbf{Y}$). In step 2, we trained a CNN to act as a digital twin of the retina. We refer to it as the forward model ($\hat{\mathscr{F}} : \mathbf{X} \rightarrow \hat{\mathbf{Y}}$). We sent the same high-resolution images to the forward model, generating a predicted response matrix ($\hat{\mathbf{Y}}$). We calculated the Poisson loss against the prediction and ground truth to update the forward model along with the regularization terms ($\theta^* = \operatorname{argmin}_\theta[\ell(\widehat{\mathscr{F}_\theta}(\mathbf{X}), \mathscr{F}(\mathbf{X})) + \mathbf{R}]$) where $\theta^*$ is the optimized parameters of $\hat{\mathscr{F}}$, $l$ is the loss function, and $\mathbf{R}$ is the regularization term. Then, we conducted hyperparameter optimization with a random search. Once the forward model is trained, we fix its weights. In step 3, we prepended another CNN, the actor network $A : \mathbf{X} \rightarrow \mathbf{X_{down}}$ where $\mathbf{X_{down}} \in \mathbb{R}^{32x32}$, which learns to downsample images. Again, we sent the same high-resolution image set into the actor network, which reduces them to lower-resolution images (32 × 32 pixels, four-fold downsampling). The low-resolution images are sent through the fixed forward model, generating a predicted response matrix $\hat{\mathbf{Y}}_{\mathbf{Actor}} = \hat{\mathscr{F}}(A(\mathbf{X}))$. Similar to step 2, we compared the predicted response of the lower-resolution images against the ground truth by calculating the Poisson loss. The loss is then used to update the actor network. The forward model remained fixed. As the actor network is updated, it learns to distill pertinent features to downsample images while generating a neuronal response similar to high-resolution images.

We determined the dimension of high-resolution image set by performing an ex-vivo experiment equivalent to the one described step 1, but with an image set containing 240 original images replicated in 4 different sizes (256 × 256, 128 × 128, 64 × 64, and 32 × 32 pixels; downsampling by pixel averaging), each projected 10 times. For each dimension, the image set was split into odd and even groups (5 repetitions each) to allow comparisons within the same image dimension. To compare between different dimensions, we always used the odd

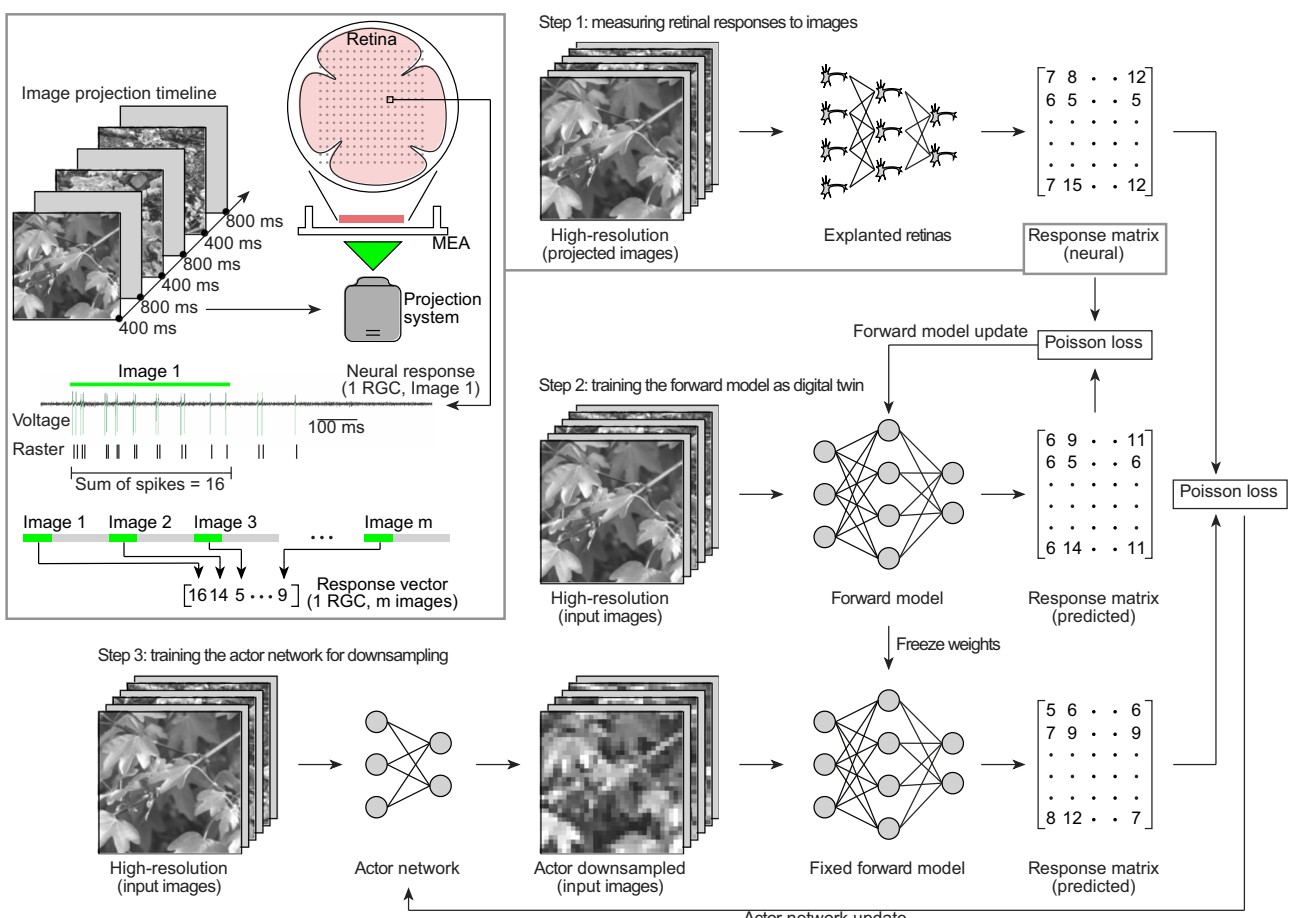

**Fig. 1 | Actor-model framework for optimal downsampling.** In step 1, the neuronal response of RGCs is measured by projecting high-resolution images onto explanted mouse retinas. Then, it is converted into a neural response matrix. The inset sketch shows how a response vector is built. Images are projected for 400 ms (green bars) separated by a 800-ms long gray frame (gray bars). RGCs are isolated after spike detection and clustering. A response vector is built for each RGC by summing up the number of spikes occurring during image projection (400-ms window). The image sequence was repeated 10 times to account for trial-to-trial variability, and responses to the same images were averaged (not shown in the sketch). Hence, numbers in the response vector are not necessarily integers. Response vectors are arranged into the response matrix. In step 2, the neuronal response matrix serves as ground truth for the training of the forward model. Given the same input, the predicted response of the model is compared against the ground truth by calculating the Poisson loss, then used to update the forward model. After training, the forward model is fixed. In step 3, the actor network is prepended to the fixed forward model. High-resolution images are passed into the actor network which downsample and feed them into the fixed forward model. The predicted response is compared against the ground truth by calculating the Poisson loss, then used to update the actor network. High-resolution image reproduced from the Open Access van Hateren Natural Image Dataset available at https://github.com/hunse/vanhateren (MIT License).

group. We recorded neural responses from identified RGCs ($n = 12$ from $N = 3$ retinas) and built corresponding response vectors. Then, we used neuronal reliability as a quantitative measure to compare the resolutions in pairs. Briefly, the ex-vivo neuronal reliability of each RGC is measured as the $R^2$ value between the neuronal responses to two paired images (Fig. 2a). Paired images are the same image presented at either the same (e.g. 256 vs 256) or different (e.g. 256 vs 128) resolutions. Pooling all the RGCs together (Fig. 2b, c), we did not find any statistically significant difference in neuronal reliability when images of 256 × 256 pixels are downsampled to 128 x 128 pixels (two-tailed paired Wilcoxon test, $p = 0.30$). For all other comparisons, we found statistically significant differences (two-tailed paired Wilcoxon tests: 256–64, $p = 0.0122$; 256–32, $p = 0.0005$; 128 – 64, $p = 0.0200$; 128 – 32, $p = 0.0005$; 64 – 32, $p = 0.0009$).

On the one hand, we aimed to minimize the high-resolution image dimension so as to reduce the number of parameters of the forward model. More parameters will result in greater computational complexity and larger risk of overfitting. Since we did not find a statistically significant difference while downsampling from 256 × 256 to 128 × 128 pixels, we rejected the 256 × 256 pixel size. On the other hand, we

wanted to maximize the high-resolution size so that it can give a statistically significant difference in neuronal reliability compared to the downsampled images, hence we chose 128 x 128 pixels for the high-resolution images. The choice of four-fold downsampling is also derived from this experiment. A reduction in neuronal reliability is required between high-resolution images and images downsampled by pixel averaging to fully leverage on the potential of the actor-model framework to downsample images with higher neuronal reliability. Four-fold downsampling (128 vs. 32) exhibited the greatest reduction in neuronal reliability (one-tailed paired Wilcoxon test, $p = 0.0002$).

## Actor-downsampled images elicit correlating responses to high-resolution images in-silico

With the forward model and actor network trained, we conducted a comparison of the performance in-silico between the actor-model framework and the pixel averaging method for four-fold downsampling (Fig. 3). Here, the forward model functions as the digital twin of an explanted mouse retina. We used different types of images as inputs (200 unique images for each group), including high-resolution images (high-resolution), images downsampled by the actor network

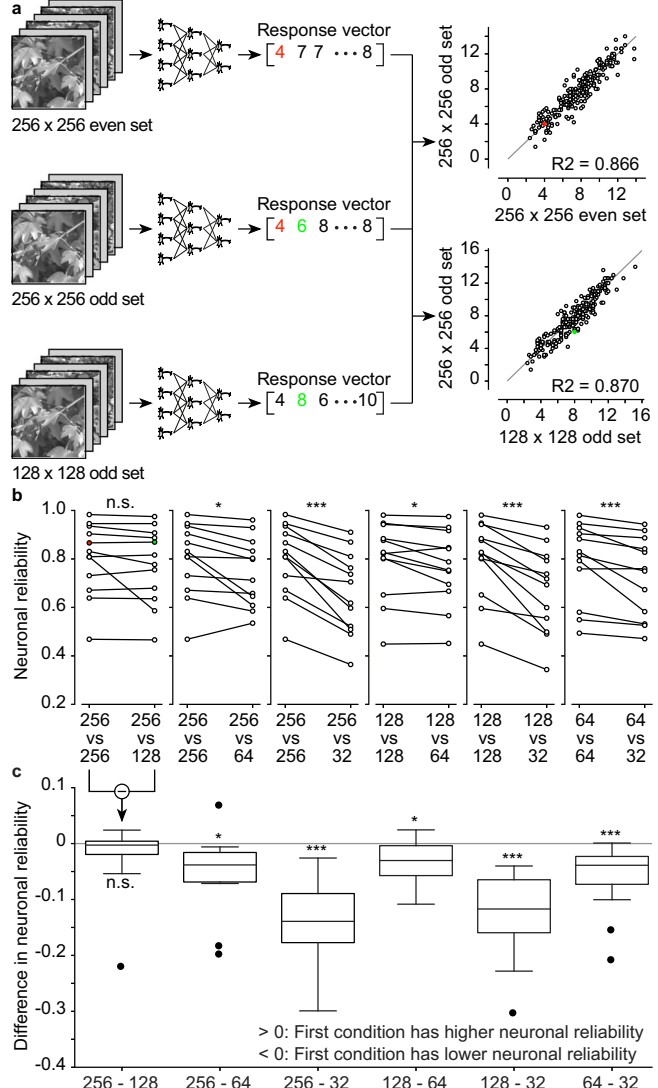

**Fig. 2 | Dimension for high-resolution images. a** Neuronal reliability of a RGC to the same image dimension (256 ×256 pixels) or to different image dimensions (256 x 256 pixels vs 128 ×128 pixels). The scatter plots are built from the response vectors. The red and green dots correspond to the red and green numbers in the response vectors. The gray lines are the identity lines. **b** Plots of the paired neuronal reliabilities for all the recorded RGCs (*n* = 12 RGCs from *N* = 3 retinas). **c**, Each boxplot is the distribution of the pairwise difference in neuronal reliability from panel **b**. The box spans from 25th to the 75th percentiles, the line is the median, and the whiskers are 1.5 times the interquartile range. The black dots indicate outliers. Two-tailed paired Wilcoxon tests: 256 − 128 (*p* = 0.30, reported as n.s.), 256 − 64 (*p* = 0.0122, reported as *), 256 − 32 (*p* = 0.0005, reported as ***), 128 − 64 (*p* = 0.0200, reported as *), 128 − 32 (*p* = 0.0005, reported as ***), 64 − 32 (*p* = 0.0009, reported as ***). Statistical analysis applies to both panels **b** and **c**. High-resolution image reproduced from the Open Access van Hateren Natural Image Dataset available at https://github.com/hunse/vanhateren (MIT License).

(actor downsampled), and images downsampled using the pixel averaging method (average downsampled). Lastly, we computed the in-silico neuronal reliability (Fig. 3a). Briefly, the neuronal reliability of each modeled neuron (*n* = 60) is measured as the $R^2$ between the neuronal responses of explanted retinas to high-resolution images (ground truth; step 1 in Fig. 1; *n* = 60 RGCs from *N* = 10 retinas) and the predicted responses of the forward model to the same high-resolution images (Fig. 3a, magenta), average downsampled images (Fig. 3a, yellow) or actor downsampled images (Fig. 3a, cyan). The comparison between predicted response of high-resolution images and ground

truth defines the baseline reliability of a modeled neuron (Fig. 3a, magenta). The ground truth compared to predicted response of downsampled images (Fig. 3a, actor in cyan and average in yellow) is the in-silico neuronal reliability for downsampled images, which is how similar the responses to downsampled images are to those from high-resolution images in-silico.

As expected, we observed a statistically significant reduction in neuronal reliability of the average downsampled images compared to the high-resolution images (Fig. 3b; two-tailed paired Wilcoxon test, *p* = 0.0001; one-tailed paired Wilcoxon test, *p* < 0.0001). However, we did not observe any statistically significant difference in neuronal reliability between high-resolution images and actor downsampled images (Fig. 3c; two-tailed paired Wilcoxon test, *p* = 0.56). Furthermore, the neuronal reliability of the actor downsampled images was significantly higher than the one of average downsampled images (Fig. 3d; two-tailed paired Wilcoxon test, *p* = 0.0001; one-tailed paired Wilcoxon test, *p* < 0.0001). It appears that some neurons exhibit low in-silico neuronal reliability, this effect could be attributed to the learning of the model. The parameters for some of the neurons could be overfitted, and since we do not model each neuron separately, it is difficult to ensure that every neuron is optimally modeled. Overall, we found a 4.9% median percent increase in neuronal reliability for actor downsampled images compared to downsampled images by pixel averaging. It is worth noticing that the performance increase is not specific to the type (ON or OFF) of the modeled neuron (Supplementary Fig. 1a; two-tailed Mann-Whitney U test, *p* = 0.3845) or to its RF size (Supplementary Fig. 1b; Pearson correlation coefficient r = 0.0713, *p* = 0.5885).

Based on these in-silico results, the actor network has found a way that can elicit a neuronal response more similar to the ground truth compared to downsampling by pixel averaging.

Among the various learning-free approaches we choose pixel averaging as the benchmark for evaluation, since it is the conventional visual encoding approach used in prosthetic devices. Nevertheless, we investigated the performance of the actor-model framework relative to other downsampling methods common in image processing. Specifically, we evaluated bilinear interpolation (Bil), nearest neighbor interpolation (Near), lanczos kernel with radius 3 (Lan3), lanczos kernel with radius 5 (Lan5), cubic interpolation (Cub), gaussian kernel (Gau), area interpolation with anti-aliased resampling (Area), Mitchell-Netravali cubic non-interpolating filter (Mit). The actor downsampled images consistently elicited higher in-silico neuronal reliability compared to the other learning-free methods (Fig. 4; two-tailed paired Wilcoxon tests: *p* = 0.0001 for Actor - Average and Actor - Area, *p* < 0.0001 for other comparisons; one-tailed paired Wilcoxon tests, *p* < 0.0001 for all). This result further accentuates the effectiveness and the necessity of having a learning-based framework tailored for the retina.

Since we did not find another downsampling method that outperforms the actor-network framework, we continued using pixel averaging as a reference method. The next logical step is to validate these in-silico findings ex-vivo in explanted retinas.

### Actor-downsampled images elicit correlating responses to high-resolution images ex-vivo

Finally, we validated the actor-model framework ex-vivo. We measured the neuronal responses of mouse retinas when presented with high-resolution images, actor downsampled images, and average downsampled images (*n* = 21 RGCs from *N* = 8 retinas; 200 unique images, 10 repeats per image). High-resolution images were presented twice: first to determine a new ground truth, and then to compute neuronal reliability of high-resolution images. We evaluated the performance of the actor-model framework by calculating its ex-vivo neuronal reliability when compared to the new ground truth (Fig. 5a). Qualitatively, results ex-vivo match in-silico

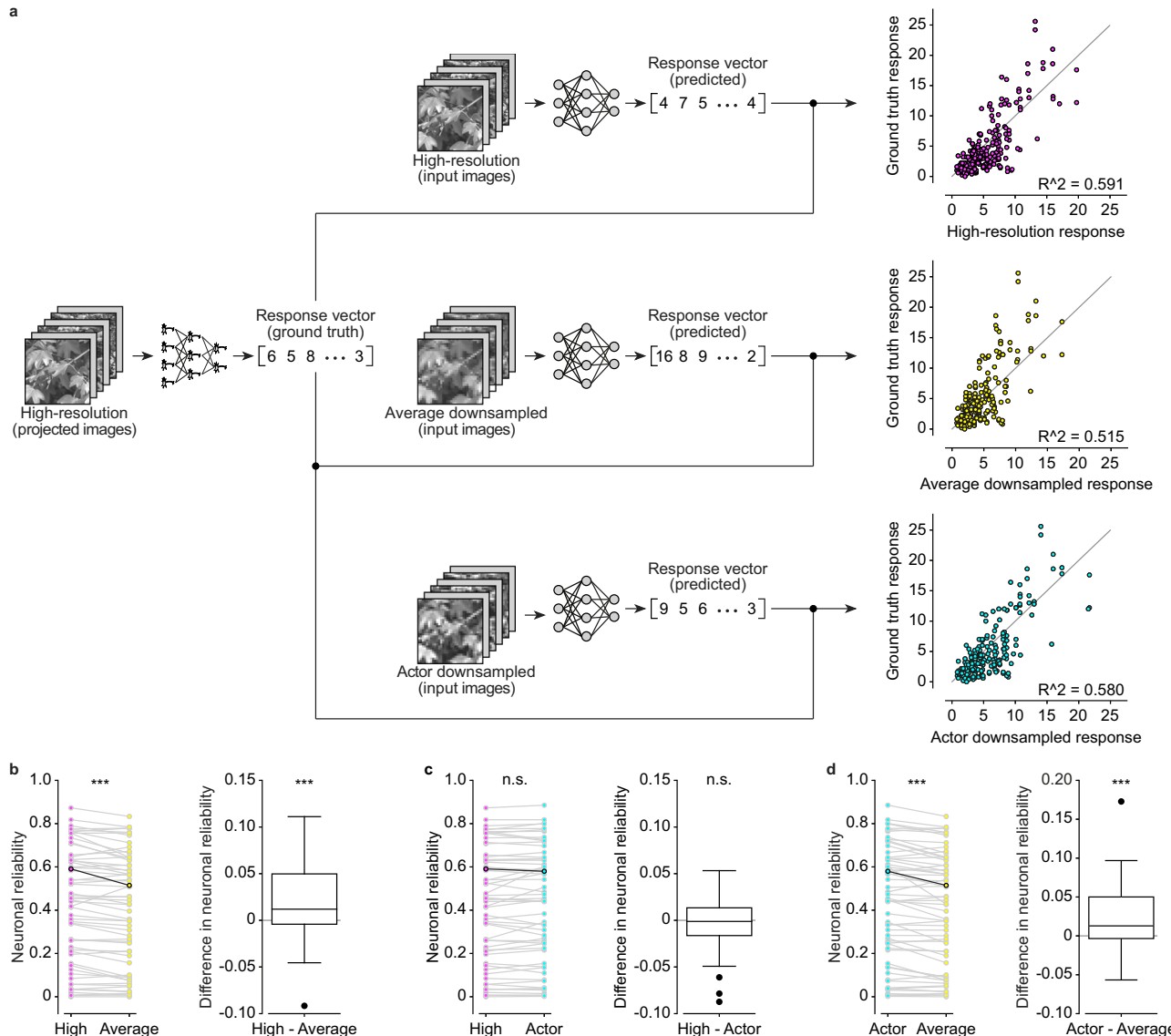

**Fig. 3 | Results in-silico for four-fold downsampling. a** High-resolution, actor downsampled and average downsample images are passed as input to the forward model to obtain a predicted response in-silico. On the right there are the scatter plots of the same representative neuron from which the $R^{2}$ value between the predicted responses and the ground truth response is computed. The gray line is the identity line. **b–d** The left plots are the paired distribution of neuronal reliabilities ($n = 60$ modeled neurons). Highlighted points correspond to the representative neuron in **a**. The boxplots show the distribution of the pairwise difference

in neuronal reliability. The box spans from the 25th to the 75th percentiles, the line is the median, and the whiskers are 1.5 times the interquartile range. The black dots indicate outliers. Two-tailed paired Wilcoxon tests: High - Average ($p = 0.0001$, reported as ***), High - Actor ($p = 0.56$, reported as n.s.), Actor - Average ($p = 0.0001$, reported as ***). High-resolution image reproduced from the Open Access van Hateren Natural Image Dataset available at https://github.com/hunse/vanhateren (MIT License).

data except for a statistically significant reduction in neuronal reliability between high-resolution images and actor downsampled images (Fig. 5c; two-tailed paired Wilcoxon test, $p < 0.0001$; one-tailed paired Wilcoxon test, $p < 0.0001$). This result was expected as information loss would occur during downsampling. The neuronal reliability of average downsampled images is still significantly lower than the one of high-resolution images (Fig. 5b; two-tailed paired Wilcoxon test, $p < 0.0001$; one-tailed paired Wilcoxon test, $p < 0.0001$). Importantly, the neuronal reliability of actor downsampled images is significantly higher than the neuronal reliability of average downsampled images (Fig. 5d; two-tailed paired Wilcoxon test, $p = 0.0012$; one-tailed paired Wilcoxon test, $p = 0.0006$). The actor downsampling method performs 2.9% better than the average downsampling method (median percentage increase). Similar to in-silico results, the performance increase is not specific

to types (ON or OFF) of RGC (Supplementary Fig. 1c; two-tailed Mann-Whitney U test, $p = 0.4003$) or to its RF (Supplementary Fig. 1d; Pearson correlation coefficient r = −0.0838, $p = 0.7181$).

It is worth reporting that we found a statistically significant difference in the average response of RGCs to high-resolution, actor downsampled and average downsampled images (Supplementary Fig. 2; Friedman test, $p < 0.0001$). In particular, the mean response to average downsampled images is significantly lower than the average response to both high-resolution and actor downsampled images (Nemenyi post-hoc test, $p = 0.001$ for both comparisons). On the contrary, the mean response of RGCs to actor downsampled images was not significantly different from the response to high-resolution images (Nemenyi post-hoc test, $p = 0.5398$). This result confirms that actor downsampled images elicited neural responses more similar to high-resolution images.

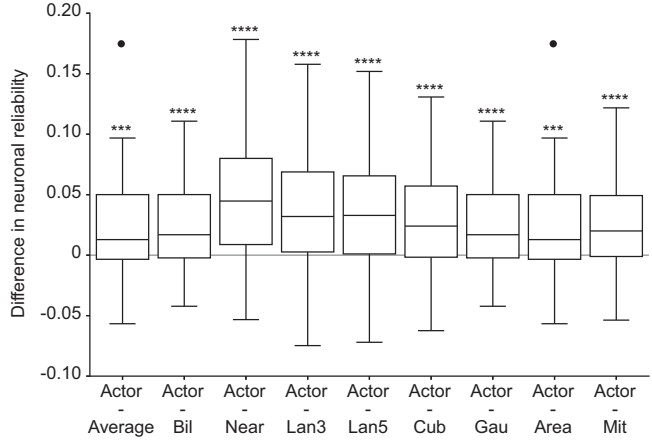

**Fig. 4 | Results in-silico for learning-free downsampling methods.** Each boxplot is the distribution of the pairwise difference in neuronal reliability between actor downsampled images and downsampled images by a learning-free method ($n = 60$ modeled neurons). The first boxplot is the same reported in Fig. 3d). The box shows the 25th and the 75th percentiles respectively, the line is the median, and the whiskers are 1.5 times the interquartile range. The black dots indicate outliers. Two-tailed paired Wilcoxon tests: Actor - Average and Actor - Area ($p = 0.0001$, reported as ***), other comparisons ($p < 0.0001$, reported as ****).

## Actor network develops features for optimal downsampling

Then, we delved deeper into the pertinent attributes of effective downsampling by comparing high-resolution images to their respective downsampled version.

Upon visual examination, it is evident that actor downsampled images better resemble the original images (Fig. 6a). Furthermore, the quantification of the local contrast difference shows that actor downsampled images exhibit a significantly higher local contrast compared to average downsampled images (Fig. 6b; two-tailed paired Wilcoxon test, $p < 0.0001$; one-tailed paired Wilcoxon test, $p < 0.0001$). This result implies that the actor network strives to increase the image contrast, highlighting its significance in generating neuronal responses akin to those of high-resolution images.

The next step is understanding how contrast affects neuronal reliability. First, we looked at the difference in local contrast of downsampled images by the other learning-free approaches (Fig. 7a; two-tailed paired Wilcoxon tests: $p = 0.8433$ for Actor – Mit, $p < 0.0001$ for all other comparisons). Most learning-free downsampling approaches preserve the local contrast better than pixel averaging, and some even better than the actor network (e.g. Near, Lan3, Lan5 and Cub; median difference in local contrast <0). This result is counterintuitive since the neuronal reliability of these methods was not higher than the actor network (Fig. 4).

To further investigate contrast, we modified the pixel averaging approach by either reducing or increasing the contrast of downsampled images before feeding them to the forward model. The gap in neuronal reliability between actor downsampled and average downsampled images widened when the contrast of average downsampled images is artificially reduced (Fig. 7b, contrast factor $\alpha < 1$). Conversely, when the contrast of average downsampled images is artificially increased ($\alpha > 1$), the gap in neuronal reliability decreases and minimizes for $\alpha = 1.5$. Moreover, when the contrast was increased further ($\alpha > 1.5$), the gap widened again. When the gap in neuronal reliability is minimized, the local contrast of the corrected average downsampled images is closer to the actor downsampled images although the difference is still statistically significant (Fig. 7c; two-tailed paired Wilcoxon test, $p < 0.0001$). This result indicates that there is an optimal level of contrast preservation which is reached by the actor network. Hence, other learning-free models fail to perform even though contrast was better preserved. Moreover, although increasing the contrast

of the average downsampled images resulted in higher neuronal reliability, the actor network still performed better (Fig. 7d; two-tailed paired Wilcoxon test, $p = 0.0082$; one-tailed paired Wilcoxon test, $p = 0.0041$).

Hence, we hypothesize that contrast might not be the only factor learned by the actor network. Indeed, the six kernels learned by the actor network closely resemble a Mexican hat shape (Laplacian of a Gaussian) which is a common function in algorithms for edge detection[40] (Fig. 8). This function, or its fast approximation difference of Gaussians, is also traditionally used to model the center-surround organization of the RGC RF[31]. This result is unexpected since the actor network was not specifically trained to mimic center-surround properties, yet it emerges when optimized based on neuronal responses. The actor-model framework could integrate edge detection into the downsampling process by convoluting with a Mexican hat function. Hence, we hypothesize that this component also contributed to a more effective downsampling of the images in conjunction with optimizing the contrasts, leading to higher neuronal reliability.

## In-silico prediction of x-fold downsampling

In the previous sections, we explored and validated the effectiveness of the actor-model framework during four-fold downsampling, where images were reduced from 128 x 128 pixels to 32 x 32 pixels. The actor-model framework elicited a higher neuronal reliability (4.9% in-silico and 2.9% ex-vivo) compared to a learning-free approach (pixel averaging). Therefore, we decided to vary the folds of downsampling in-silico to understand the extent of aggregation before the information loss cannot be recovered by the framework (Fig. 9). We trained multiple actor networks as described in the preceding sections (960 unique images) to downsample the high-resolution images, each learning a different downsampling fold: 2-fold (64 x 64 pixels), 4-fold (32 x 32 pixels), 8-fold (16 x 16 pixels), 16-fold (8 x 8 pixels), and 32-fold (4 x 4 pixels). We observe statistically significant differences in neuronal reliability between actor downsampled images and average downsampled images up to 8-fold downsampling ($n = 60$; two-tailed paired Wilcoxon tests: 2-fold, $p = 0.0178$; 4-fold, $p = 0.0004$; 8-fold, $p = 0.0006$). However, from 16-fold onwards, this difference was not observed ($n = 60$; two-tailed paired Wilcoxon tests: 16-fold, $p = 0.8024$; 32-fold, $p = 0.4483$;). This was expected as 16-fold downsampling corresponds to reducing the original image from 128 x 128 pixels to 8 x 8 pixels. Therefore, each pixel has a size of 400 x 400 $\mu m^2$, exceeding the RF size of most RGC in the mouse retina[41] (Supplementary Fig. 1). Hence, the amount of information loss during downsampling might be too much to be recovered.

Although we observed a significant difference between the neuronal reliability of actor downsampled images and average downsampled images for 2-fold downsampling, a significant difference was not observed between neuronal reliability of average downsampled images to high-resolution images (two-tailed paired Wilcoxon test, $p = 0.1925$), while actor downsampled images has slightly different neuronal reliability than high-resolution images (two-tailed paired Wilcoxon test, $p = 0.0277$; one-tailed paired Wilcoxon test, $p = 0.0139$). This result suggests that the forward model is robust to downsampling. This is not out of expectation as we used a CNN to model the retina, which could possess a certain level of robustness to downsampling.

## Discussion

In this study, we have illustrated the efficacy of an actor-model framework to learn a downsampling method that outperforms common learning-free techniques. Furthermore, we substantiated the effectiveness of our approach by analyzing the neuronal responses of RGCs, and we identified contrast as a crucial feature for effective downsampling. Additionally, we observed the emergence of distinct

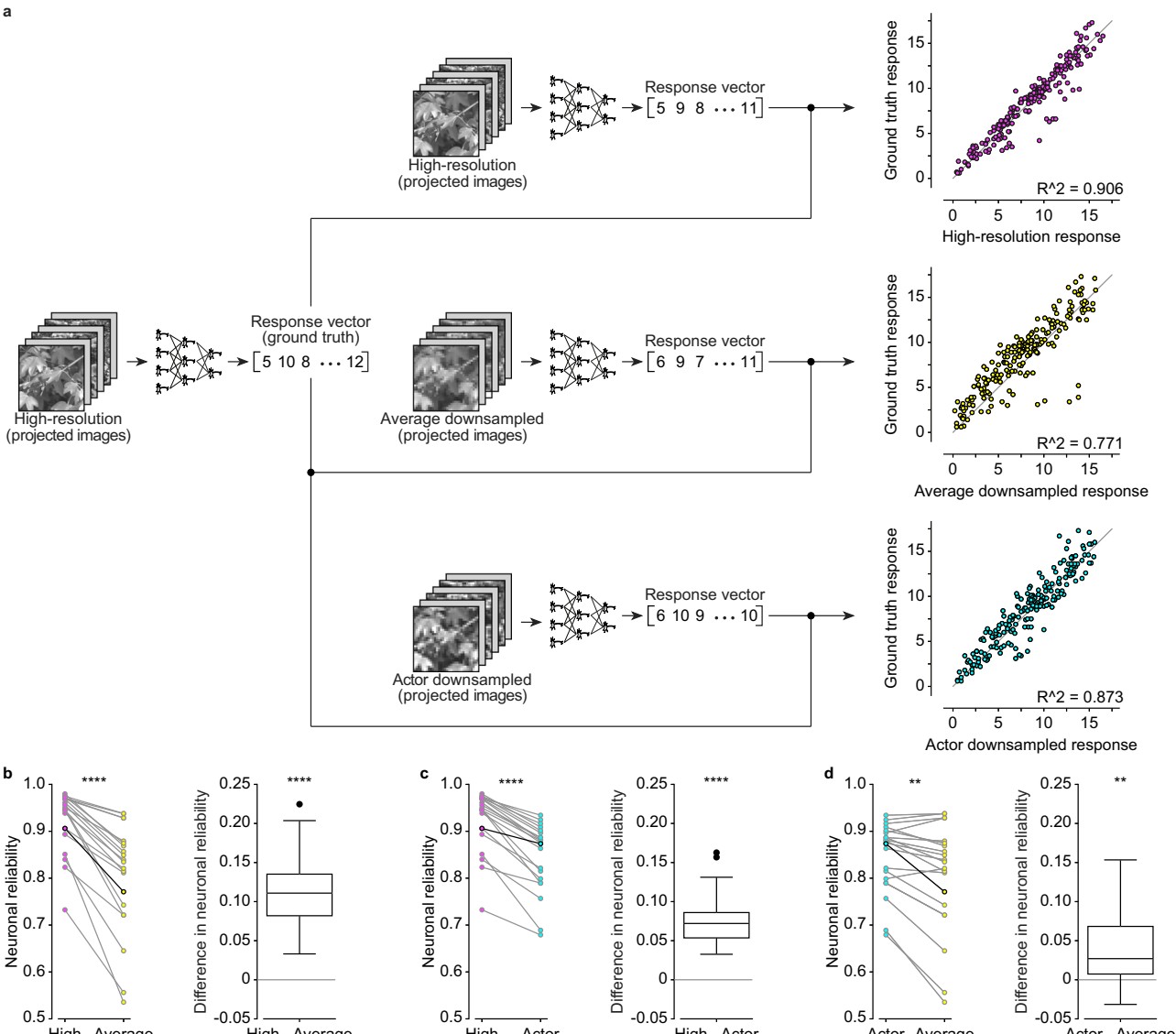

**Fig. 5 | Results ex-vivo for four-fold downsampling. a** High-resolution, actor downsampled and average downsample images are projected onto retinas explanted over a MEA. On the right there are the scatter plots of the same RGC from which the R² value between the obtained responses and the ground truth response is computed. The gray line is the identity line. **b–d** The left plots are the paired distribution of neuronal reliabilities for all the RGCs (n = 21 RGCs from N = 8 retinas). Highlighted points correspond to the RGC in **a**. The boxplots show the

distribution of the pairwise difference in neuronal reliability. The box spans from the 25th to the 75th percentiles, the line is the median, and the whiskers are 1.5 times the interquartile range. The black dots indicate outliers. Two-tailed paired Wilcoxon tests: High vs Average (p < 0.0001, reported as ****), High vs Actor (p < 0.0001, reported as ****), Actor vs Average (p = 0.0012, reported as **). High-resolution image reproduced from the Open Access van Hateren Natural Image Dataset available at https://github.com/hunse/vanhateren (MIT License).

patterns in the learned kernels of the actor network, mimicking the center-surround properties of RGCs. In this section, we assess our approach against the state-of-the-art in the field, we delve into the implications of our findings, and propose potential avenues for advancing image encoding research.

Previously, the actor-model framework was employed within a physical system, where it successfully learned the necessary input to yield any desired output when transmitted through a multimode fiber. The performance achieved in that study was comparable to gold-standard methods[37]. Here, we sought to harness the potential of the actor-model framework within a biological system, which is often characterized by complexity and high variability. In contrast to a physical system, we encountered a distinct set of challenges during the experimentation process. Firstly, like in many biological ex-vivo experiments, the difficulty lay in the sample viability, which constrained the duration of data collection. Given that neural networks

typically require vast amounts of data to be effective, this limitation poses a significant obstacle to training the neural network efficiently. Secondly, the number of neurons recorded could fluctuate depending on the quality of the dissected tissue and other factors. Although we used 256 electrodes, not every electrode could successfully capture the electrical activity of RGCs. These issues imposed conducting experiments on multiple retinas and consolidating the recorded RGCs into a single dataset. As such, the framework must learn and generalize across different mice to capture inter-sample variability, which could add another layer of difficulty. Lastly, intrinsic variability within the retina presented another hurdle. The response could vary from trial to trial when the same image was projected twice, with this variability being more pronounced than in the physical systems used in the previous work. Consequently, our actor-model framework required greater robustness to be effective. Nevertheless, the actor-model framework successfully learned efficient downsampling. This

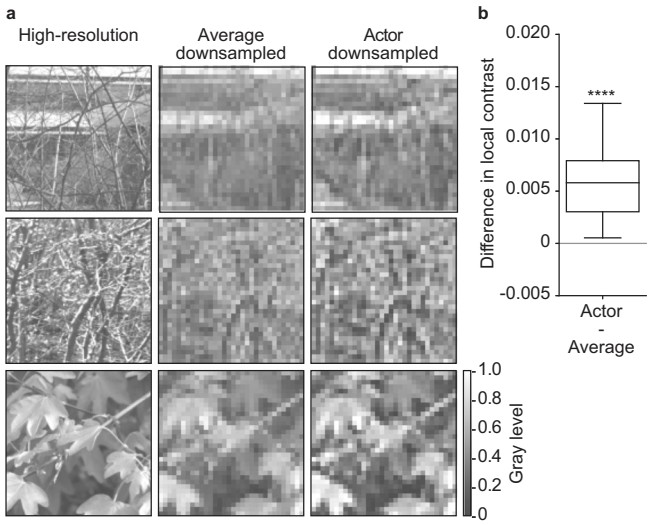

**Fig. 6 | Actor network preserves contrast. a** On the left, there are three high-resolution images. The same images after four-fold average downsampling (middle) and four-fold actor downsampling (right). **b** Distribution of the pairwise difference in local contrast between actor and average downsampled images ($n = 200$ unique images; Two-tailed paired Wilcoxon test, $p < 0.0001$, reported as ****). The box spans from the 25th to the 75th percentiles, the line is the median, and the whiskers are 1.5 times the interquartile range. High-resolution images reproduced from the Open Access van Hateren Natural Image Dataset available at https://github.com/hunse/vanhateren (MIT License).

achievement underscores the robustness of the actor-model framework and its capacity to discover solutions irrespective of whether it is applied to a physical or biological system. We contend that the potential of the actor-model framework is not limited to image encoding in the retina, and we look forward to future experiments that could capitalize on this versatile framework. For example, in the context of tactile feedback, Eldeeb and Akcakaya demonstrated the potential of using electroencephalography (EEG) to guide the electrical stimulation parameters[42]. The actor-model framework could also be applied in this situation, where the actor network could determine the electrical stimulation required to elicit the desired EEG response.

In the actor-model framework, artificial image encoding depends on two components: the forward model (digital twin of the retina) and the actor network (artificial image encoding). Much research has been previously dedicated to deriving a forward model of the retina[32,35,36,43–47]. Selecting a suitable forward model plays a crucial role in this study. Conventional models such as linear-nonlinear model and generalized model have been shown to model the retina. However, such conventional models generally fail to represent retinal responses to natural scenes[35]. On the other hand, CNNs have been shown in recent years to be significantly better in modeling the retina for both white noise and natural scenes[35,36], suggesting a level of complexity hidden within its seemingly simple structure. Hence, we constructed a digital twin of the retina using a CNN forward model which accounted for the inherent computation of the retinal network. In this study, the innovation lies in the combination of a forward model with a CNN encoder (actor network), while the architecture of the forward model is derived from the state-of-the-art[35]. Since the actor network optimized the image encoding process based on the output of the forward model, the positive results observed in this study also indicate the robustness of the forward model per se. Hence, our results highlight the suitability of CNNs as a forward model of the retina. However, as the field progresses and more forward models are derived, it will be possible to substitute the forward model used in this study with a more accurate one[48].

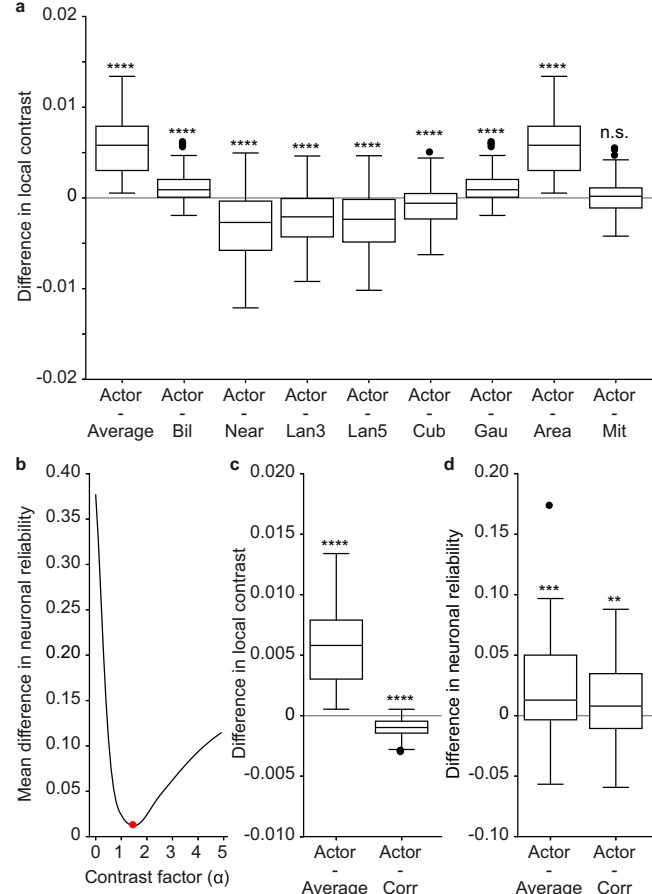

**Fig. 7 | Effect of contrast on neuronal reliability. a** Each boxplot is the distribution of the pairwise difference in local contrast between actor downsampled images and downsampled images by a learning-free method ($n = 200$ unique images). Two-tailed paired Wilcoxon tests: Actor - Mit ($p = 0.8433$, reported as n.s.), other comparisons ($p < 0.0001$, reported as ****). The first boxplot is the difference between actor downsampled and downsampled images by pixel averaging (as in Fig. 6b). **b** Mean difference in neuronal reliability between actor downsampled images and downsampled images by pixel averaging with corrected contrast as a function of the contrast factor $\alpha$ (average of $n = 60$ modeled neurons). The red circle indicates the minimum of the curve for $\alpha = 1.5$. **c** Distribution of the pairwise differences in local contrast ($n = 200$ unique images) between actor downsampled images and images downsampled by pixel averaging (left, as in Fig. 6b) and images downsampled by pixel averaging with optimized contrast at $\alpha = 1.5$ (right). Two-tailed paired Wilcoxon tests: Actor - Average ($p < 0.0001$, reported as ****), Actor - Corr ($p < 0.0001$, reported as ****). **d** Distribution of the pairwise differences in neuronal reliability ($n = 60$ modeled neurons) between actor downsampled images and images downsampled by pixel averaging (left, as in Fig. 4) and images downsampled by pixel averaging with optimized contrast at $\alpha = 1.5$ (right). Two-tailed paired Wilcoxon tests: Actor - Average ($p = 0.0001$, reported as ***), Actor - Corr ($p = 0.0082$, reported as **). The boxes span from the 25th to the 75th percentiles, the line is the median, and the whiskers are 1.5 times the interquartile range.

As visual technology progresses and electrode resolution increases, image encoding methods become increasingly crucial. This study demonstrates that a learning-based approach, which accounts for the biological retinal process, yields superior results. We foresee this learning-based method serving as a foundation for future research, potentially leading to the identification of the most effective image encoding technique. This study shows the benefits of using CNNs for image encoding (actor network) which, to the best of our knowledge, their use for efficient image encoding has not been reported despite their great potential. CNNs are efficient in capturing the relevant

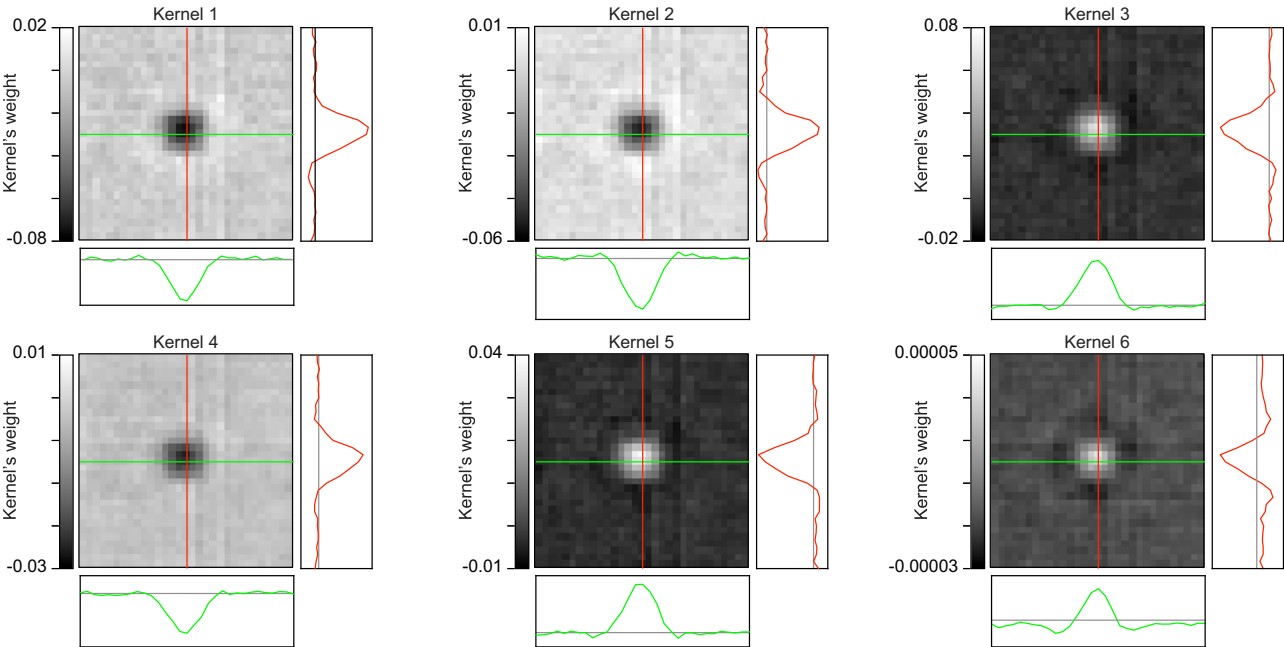

**Fig. 8 | CNN learned kernels.** Visualization of the six kernels learned by the actor network. For each plot, the green profile corresponds to the line plot of the kernel across the horizontal midline (in green). The red profile corresponds to the line plot of the kernel across the vertical midline (in red). The kernels are categorized into two distinct groups: those with predominantly positive weights and those with predominantly negative weights, resembling center ON and center OFF properties of RGCs.

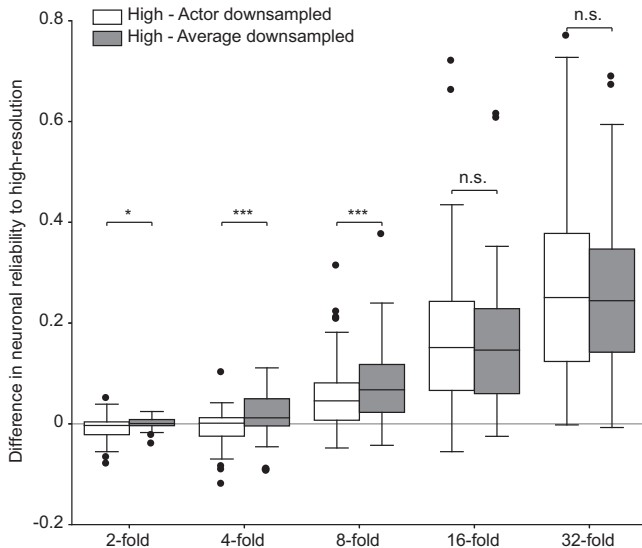

**Fig. 9 | In-silico difference in neuronal reliability compared to high-resolution images (baseline) for x-fold downsampling.** Each boxplot is the distribution of the pairwise difference in in-silico neuronal reliability between the downsampled images (actor in white and average in gray) and high-resolution images for different downsampling folds ($n = 60$ modeled neurons). The box spans from the 25th to the 75th percentiles, the line is the median, and the whiskers are 1.5 times the interquartile range. The black dots indicate outliers. Two-tailed paired Wilcoxon tests: 2-fold ($p = 0.0178$, reported as *), 4-fold ($p = 0.0004$, reported as ***), 8-fold ($p = 0.0006$, reported as ***), 16-fold ($p = 0.8024$, reported as n.s.), 32-fold ($p = 0.4483$, reported as n.s.).

features of the image. This model highlighted the importance of contrast during downsampling, confirming also its importance for a CNN forward model of the retina[35]. In general, employing a learning-based approach enables further analysis of the learned features, providing insights into the underlying dynamics of the system. Also, the

weight-sharing properties of CNNs allow for generalization within each retina and across different retinas. Weight-sharing property in CNNs refers to how each learned kernel is applied across the entire image, thus reducing the number of parameters to be learned. A significant challenge in biological experiments involves conducting multiple trials with different samples. RGCs are recorded from multiple sessions and combined into a single dataset. However, even with such methods, it is not guaranteed that every area of the image is captured by the neuronal activity. Nevertheless, since weights are shared within a CNN, the actor network can extract pertinent features based solely on the subset of RGCs measured and subsequently apply the learned kernels through the whole visual space. These distilled features can then generalize to other retinas and to other areas of the images not captured by neurons, as demonstrated by the results obtained in the ex-vivo experiments for validation. Although we could not guarantee that the RGCs recorded in the ex-vivo validation experiment (Fig. 5) would co-locate with the RGCs used for the forward model and actor network training (Fig. 1), the actor network generalization capability highlights the robustness and invariance towards the location of the RGCs recorded. Additionally, we show that the actor network produces images that are neither biased towards the type of RGCs nor to their RF size (Supplementary Fig. 1).

Several visual encoding approaches have been proposed so far. In computer vision, a conventional approach is based on saliency detection (e.g. edge detection, object detection or content-aware retargeting). These approaches generally aim at reducing the complexity of the image, highlighting interesting content and features while removing less interesting information[27,28,49,50]. The actor-model framework is compatible with saliency-based detection. First, one would apply saliency detection algorithms to extract important information, and then pass the processed images through the actor-model framework for more effective downsampling. In visual prostheses, image encoding can consist of two phases: image processing (e.g. downsampling) followed by stimulus optimization. Most of the image encoding strategies focus on the stimulus optimization rather than on the image processing phase, which is typically based on pixel

averaging when present. A 'naive' approach for stimulus optimization sets the stimulation strength of each electrode based on the corresponding pixel value of the downsampled images[51].

At the biophysical level, Ghaffari et al. utilized a feedforward neural network to optimize the stimulation parameters to achieve a more localized RGC activation but validated it only in-silico[52]. Similarly, Spencer et al. tried to approximate and invert a biophysical forward model that converts the stimulation strength to retinal activation[53]. Both studies aim to produce a more focused retinal activation and thus could potentially be integrated with the actor-model framework for the next phase which involves the conversion of the more effective actor downsampled images to stimulation parameters.

At the perceptual level, Fauvel et al. presented a preferential Bayesian optimization algorithm to optimize a visual encoder based on patients' feedback[54]. However, the encoder was not proposed, and the optimization was based on a prosthetic vision simulator which generates patterns of phosphenes (phosphene model) mimicking the visual experience of patients implanted with the Argus II® device[29]. Shah et al. proposed a greedy iteration to select electrical stimuli that minimizes the error between the expected visual perception and the target image[55]. The model for the expected perception was obtained by projecting continuous white noise and measuring the RGC responses. Then, a linear filter is fitted for each RGC to find the reconstruction matrix. They perform a closed loop optimization of the electrical stimuli. First, they provided electrical stimulation and measured the RGC response within a temporal window. Then, using the reconstruction matrix, they predicted the perception and used it as the loss function for the optimization. One of the limitations of this approach is the reliance on the expected perception. Since the expected perception is obtained by the reconstruction matrix, it is unclear how it could be derived for a blind patient. In our study, we faced a similar issue since both the actor and the model networks are optimized to a set of RGCs. However, we showed in the validation experiments that the actor-model framework can generalize across retinas.

A few other studies have tried to optimize stimulation parameters using neural networks[56–58]. Van Steveninck et al. proposed an end-to-end optimization strategy in conjunction with a fixed phosphene model[56]. They suggested mapping the image to stimulation parameters using a CNN (encoder). However, they also included a secondary CNN (decoder) to post-process the output of the phosphene model, reconstructing the image. Throughout this process, both the encoder and the decoder are trained concurrently by minimizing the loss function between the reconstructed and the original image. Thus, it is unclear if the encoder had genuinely learned the optimal stimulation parameters, or the decoder is simply adept at reconstructing the image. Relic et al. addressed this issue by using a neural network surrogate of the phosphene model and removing the secondary decoder[57]. In this case, since the phosphene model is non-differentiable, a surrogate model which consists of a neural network is needed as replacement so that the loss value could propagate backwards to train the encoder model to produce the optimal stimulation parameters. Granley et al. refined this pipeline by deriving a differentiable phosphene model, removing the need of a surrogate model[58].

These studies methodologically relate to our study since CNNs were used. However, one key difference lies in the scope. Our actor-model framework optimizes the image encoding process by learning the most efficient downsampling pipeline while their method optimizes the electrical stimulation parameters directly. As a result, the actor-model framework could easily translate clinically as it improves on the conventional method which involves downsampling the images, and the features pertinent to effective downsampling found in this study can generalize well to different patients. On the other hand, the studies of Relic et al. and Granley et al. were performed in-silico only and the phosphene model used to predict the perceived perception

had to be fitted to each individual patient. In both studies, the network architecture can be decomposed into encoder and decoder, where the encoder learns to generate the optimal electrical stimulation parameters while the decoder is either a differentiable or surrogate phosphene model aiming at predicting the perceived phosphenes. This is analogous to our actor network which learns to optimally downsample the images and our model network which acts as the digital twin of the retina. While these studies bear resemblances to our work, a shared theme across them is the application of phosphene models for in-silico performance measures without validation. On the contrary, the actor-model framework showed better performance in both in-silico and ex-vivo environments using neuronal reliability as performance index.

As a step toward a learning-based approach for image encoding, there remains ample room for improvement and exploration. In this study, we projected static images and summed the number of spikes within a specified window. Constrained by the hardware employed, we were unable to present images in a continuous format (i.e., movie format), which prevented us from verifying whether our proposed methods would be applicable to more dynamic natural scenes. A logical next step would be to validate these results using continuous projections of natural scenes. In addition to exploring continuous format, the same approach could be evaluated on reducing the bit size of the depth of the images concurrently while downsampling. Reducing from 256 levels of grayscale to 8 levels would also be useful for visual prostheses to better calibrate the strength of electrical stimulation.

Another avenue is investigating the generalization of our approach across species. The actor-model framework was trained using data collected from mouse retinas. We also discovered that the results derived from these trained models could be generalized across different mouse retinas, as the validation experiments were conducted with new retinas. However, mouse retinas have anatomical and functional differences from human retinas. It would be interesting to explore whether projecting the various downsampled images onto retinas of other species would yield similar improvements (e.g. non-human primates). If successful, this would imply that the features learned by the actor network may possess the capacity to generalize even across different species, highlighting the potential for broader applicability of this method, in particular towards visual prostheses in humans.

In conclusion, this study presents a neural network-based approach for optimizing image distillation in the context of visual prostheses. The proposed actor-model framework learns to downsample images while accounting for the biological processes of the retina, resulting in more effective downsampling patterns. This research not only contributes to the advancement of image encoding techniques for visual prostheses but also highlights the importance of incorporating natural biological transformations. In general, integrating neural networks into sensory encoding could hold the key to better perception of the visual prosthesis users. Future research could build upon this learning-based approach to develop even more accurate and effective image encoding methods, enhancing the quality of life for individuals relying on such devices.

## Methods

### Electrophysiological recordings

Animal experiments were authorized by the Direction Générale de la Santé de la République et Canton de Genève in Switzerland (authorization number GE31/20). C57BL/6J mice ($n = 21$; age $75.3 \pm 26.4$ days, mean $\pm$ SD) were dark-adapted for 1 hr prior to euthanasia. Euthanasia was carried out via intraperitoneal injection of sodium pentobarbital (150 mg kg$^{-1}$). Dissection and recording of retinas were performed in carboxygenated (95% $O_2$ and 5% $CO_2$) Ames medium (USBiological, A1372-25) under dim red light. Retinas were maintained at 25 °C throughout the experiment. Explanted

retinas were positioned on a poly-lysine coated membrane (Sigma, P8920; Repligen 132544), with the RGCs side facing a 256-channel MEA (256MEA200/30iR-ITO, Multichannel systems) with 30-μm electrodes spaced 200 μm. The data sampling rate was set at 25 kHz. Data were collected using a custom python code.

## Visual stimulation

Images were projected using a custom-built setup with a Digital Mirror Device (V-7000 Hi-Speed V-Modules, ViALUX) coupled to a white LED (MWWHF2, Thorlabs). The stimulus was focused on the photoreceptors via standard optics, with an average power of 13 nW. The projected area covered 3.2 x 3.2 mm$^2$. The image set is the Open Access van Hateren Natural Image Dataset (available at https://github.com/hunse/vanhateren, MIT License), comprising 4212 monochromatic and calibrated images captured in a variety of natural environments[59], further processed to maintain a linear relationship between scene luminance and pixel values[35]. This processing step was described as crucial to prevent the retinal system from having to adapt to varying light intensity levels found in different environments[35]. The final image set used in this experiment (3190 images) was obtained from Goldin et al.[35], which was then sub-sampled for the different experiments conducted. Each image is projected for 400 ms, and an 800-ms long gray frame is inserted between images to return the firing rate to baseline value.

## Spike detection and sorting

The SpyKING CIRCUS algorithm (version 1.1.0) was used for spike sorting[60]. Manual inspection was performed using Phy software (version 2.0b5)[61], including verification of gaussian distribution in the amplitudes, waveforms present in multiple channels, presence of a dip in autocorrelogram, and merging/separating clusters as necessary. To assess the reliability of the recorded neurons and account for experimental drift, random binary checkerboard stimuli was presented at the start of the experiment, and then redisplayed roughly every half an hour[43]. The check size was 50 μm, the refresh rate was 33 Hz and the presentation time was 5 min. The correlation coefficient of a cell's average response to the same stimulus across different blocks of trials was calculated. Only neurons with a correlation exceeding 0.3 were selected for further analysis[36]. Consequently, many neurons that were not responsive over the entire experimental period (approximately 5 hrs) were discarded. To filter out poor-quality clusters, we presented a random binary checkerboard at the end of the experiment for 1 h to characterize the RF size of each spike-sorted RGC by spike-triggered averaging (STA) analysis. STA is given by $STA = \frac{1}{n_{sp}}\sum_{i=1}^{T} y_i \mathbf{X}_i$ where $n_{sp} = \sum_{i=1}^{T} y_i$ is the total number of spikes, $y_i$ is the spike count in the bin and $\mathbf{X}_i$ is the binary checkerboard at $i$ time bin. Clusters not displaying a recognizable RF were excluded from further analysis.

## Image analysis

Image analysis was performed in Python (version 3.10.6) with the OpenCV library (version 4.6.0.66). Local contrast of every downsampled image was quantified by averaging the luminance variance of a sliding window of size 7 pixels x 7 pixels. The RF size of RGCs was measured after STA analysis. The background noise was smoothed by convoluting a median filter and the contrast was enhanced to better separate the RF from the background. Last, a 2D Gaussian was fitted to measure the RF area. The average model with enhanced/reduced contrast (corrected) was obtained by the TensorFlow method accordion to $X_{corr} = [\mathbf{X}_{down} - \bar{\mathbf{X}}_{down}]^*\alpha + \bar{\mathbf{X}}_{down}$ where $\mathbf{X}_{down}$ are the downsampled images, $\bar{\mathbf{X}}_{down}$ is the average downsampled image and α is the contrast factor.

## Model architecture

The forward model architecture adheres to the current state of the art[35,62,63]. It consists of two layers, the first layer of the CNN, $k_{rsk}$, convolutes the input image. The output of the first convolution layer passes through a pointwise nonlinear function, $f_{\theta_{k[1]}}$, to obtain non-negative activation values. For each neuron n, a readout weight $w_{ijkn}$, which factorizes as $w_{ijkn} = u_{ijn}v_{kn}$, is applied. Here, $i$ and $j$ index space, with $u_{ijn}$ representing spatial weights and $v_{kn}$ denoting feature weights. Another nonlinear activation function, $f_{\theta_{k[2]}}$, is performed, followed by the utilization of a Poisson noise model during training. Softplus was chosen as the activation function for $f_{\theta_{k[1]}}$ and $f_{\theta_{k[2]}}$. Essentially, the k$^{th}$ neuron of the first layer is represented as $A_k = f_{\theta_{k[1]}}(\mathbf{K}_k{}^*\mathbf{X})$. The spiking rate $r_n$ of the n$^{th}$ neuron given an input image $\mathbf{X}$ was $r_n(\mathbf{X}) = f_{\theta_{n[2]}}(\sum_k \sum_{ij} u_{ijn}v_{kn}A_{ijk})$. Additionally, batch normalization was applied to the outputs of the first layer. Laplacian regularization was applied to the convolutional kernels of the first layer. L1 regularization was applied to the convolutional kernels of the first layer. For the feature and spatial weights of the second layer, we used

L1 regularization such that: $L_\Delta = \frac{\sum_k \sum_{rs}(\mathbf{K}_{rsk}{}^*\Delta)2_{rs}}{\varepsilon + \sum_{rsk}(\mathbf{K}_{rsk})^2}$ and $L_{l1} = \lambda_{sp}\sum_{ijn}|u_{ijn}| + \lambda_f \sum_{kn}|v_{kn}|$ with $\varepsilon = 10^{-8}$. The actor network consists of a convolution kernel that was prepended to the forward model (Supplementary Fig. 3). Poisson loss was used for the optimization of the actor network.

## Forward model and actor network training

Given m image-response pairs $(\mathbf{X}_1, \mathbf{y}_1),...,(\mathbf{X}_m, \mathbf{y}_m)$, the loss function of the forward model is provided by $L = (\frac{1}{m}\sum_{k=1}^{m} r(\mathbf{X}_k) - y_k \log r(\mathbf{X}_k)) + L_{l1}\lambda_{l1} + \lambda_\Delta L_\Delta$. The first term corresponds to the Poisson loss, while the second and third terms represent the regularization terms. The model was fitted with the Adam optimizer on the training set (1200 images, 0.8:0.1:0.1 train:validation:test split). The responses were obtained by summing up the number of events elicited within a 400-ms window from the image onset. The training of the actor network was similar except we only used L2 regularization instead of the second and third term used in the training of the forward model. We maintained a constant batch size of 32 during training for both the actor network and the forward model. For the learning rate, we started with 0.001 for forward model and 0.002 for actor network. To avoid overfitting, we employed both early stopping and the decay mechanism with maximum 1000 epochs. The hyperparameters for the regularization term were optimized by performing a random search for the forward model, while a grid search was employed for the actor network. The optimal hyperparameter values were the ones whose model produced the lowest loss value without regularization terms on the validation dataset. The hyperparameters for the forward model were 0.0033 for smoothing factor of convolution kernels, 0.00278 for spatial sparsity factor and 1.34$^{-6}$ for feature sparsity factor. For the actor network, the best run had 6 convolution kernels of size 31 x 31 with 0.1 L2 regularization.

## Neuronal reliability

As a quantitative measurement of performance for both in-silico and ex-vivo experiments, we calculated the R$^2$ value. For each neuron, we calculated the R$^2$ value for each input image and found the average across the images and neurons, such that: $R = \frac{\sum(\mathbf{x}_i - \bar{\mathbf{x}})(\mathbf{y}_i - \bar{\mathbf{y}})}{\sqrt{\sum(\mathbf{x}_i - \bar{\mathbf{x}})^2(\mathbf{y}_i - \bar{\mathbf{y}})^2}}$. Where $\mathbf{x}$ and $\mathbf{y}$ can be the predicted response or actual neuronal response to different sets of images.

## Statistical analysis

Statistical analyses were conducted with the Python Scipy library (python version 3.10.6, scipy version 1.9.1). The Shapiro-Wilk normality test was performed to justify the use of non-parametric tests.

## Reporting summary

Further information on research design is available in the Nature Portfolio Reporting Summary linked to this article.

## Data availability

The authors declare that the data supporting the findings of this study are available in the paper. The dataset to replicate the study is available at https://doi.org/10.5281/zenodo.10515301. The image set used in the study was obtained by Dr. Olivier Marre (https://doi.org/10.1038/s41467-022-33242-8). The original image set is the Open Access van Hateren Natural Image Dataset available at https://github.com/hunse/vanhateren (MIT license). Any additional requests for information can be directed to, and will be fulfilled by, the corresponding author. Source data are provided with this paper.

## Code availability

The code used in this study is available at https://doi.org/10.5281/zenodo.10519578.

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

## Acknowledgements

We would like to thank Prof. Silvestro Micera (EPFL, Switzerland) for reading the manuscript and providing valuable feedback. We would also like to thank the team of Dr. Olivier Marre (Institut de la Vision, France) for providing the code and preprocessed images used in this manuscript and the team of Dr. Serge Picaud (Institut de la Vision, France), for their advice to improve the setup for retinal recording. The project was supported by the EPFL STI e-seed fund.

## Author contributions

F.L. performed experiments, modeled and analyzed data and wrote the manuscript. B.R. performed experiments. D.P. and C.M. designed the study. D.G. designed and led the study and wrote the manuscript. All the authors read and edited the manuscript.

## Competing interests

The authors declare no competing interests.

## Additional information

**Peer review information** : *Nature Communications* thanks Maxwell Turner, Al-Azawi Mohammad and the other anonymous reviewers for their contribution to the peer review of this work. A peer review file is available.

