## [Peer Review File · Nature Communications]

REVIEWER COMMENTS

Reviewer #1 (Remarks to the Author):

In this manuscript, Leong and colleagues describe a method to learn a down-sampling encoding of natural images that preserves information important in retinal processing. They use a convolutional neural network model to predict RGC responses to flashed natural images, and then use this trained model to learn a second, downsampling network that minimizes changes in neural responses for a given downsampling factor. They then find that the learned downsampling network outperforms average downsampling in terms of neuronal reliability, and that what the learned downsampling seems to be doing is better preserving the contrast in the image compared to average downsampling.

The actor-model framework used here is very interesting and the biggest strength of the paper. The weaknesses of the paper include: a lack of clarity around what exactly the learned downsampling is doing, and whether it is actually better than a learning-free downsampling that preserves image contrast; and experimental concerns like lack of detail about cell types and small numbers of neurons per electrode recording. These weaknesses can be addressed with some *in silico* experiments, simple analyses, and small changes to data presentation and writing. Overall, this manuscript describes a very interesting framework for probing sensory encoding.

Major issues

1. My first set of suggestions has to do with the question of what the learned downsampling is doing to the images. The finding that the learned downsampling was effectively enhancing the contrast of the image relative to the average downsampling method was interesting, but it raises the question: is this all that the learned downsampling is doing, or is there more to it than that?
 - a. The authors should compare their learned downsampling to some unlearned downsampling methods that maintain the contrast of the original image. Using the model should be sufficient, and presenting new stimuli to *ex vivo* retinas is not necessary.
 - b. The authors should measure the effect on local contrast of the other unlearned downsampling methods in Fig. 4.
 - c. The responses to average downsampled images (both *in silico* and *ex vivo*) seem to generally be weaker than responses to high-resolution or actor down-sampled images. This is consistent with the idea that average downsampling is just lowering the contrast, leading to weaker responses across all images.

The authors should test whether this difference in overall response strength is present, and modify the presentation of the data to make this more clear. For example, the far right scatter plots in 3a and 5a do not have X & Y axis labels that show which axis is which condition. I also suggest inserting a line of unity on these plots to show whether a condition drives higher or lower responses.

d. If all that the actor down-sampling is doing is rescaling the contrast to match that of the original image, it leads one to wonder how this downsampling method would help in neural prosthesis design. For a prosthetic device, the conversion from contrast in the image to electrical stimulation is arbitrary anyway, so could you not achieve a similar improvement by just upping the gain from sensor to stimulator to achieve a few more spikes per unit contrast? The authors should give a description of either: 1. How the actor downsampling is doing something more interesting than a uniform rescaling of response gain or 2. How such a rescaling would assist in the design of a prosthetic device.

2. Is the learned downsampling dependent on the predictive power of the forward model? It is important to know whether we can expect improvements in forward modeling to yield changes in learned downsampling methods. The authors state that the CNN was chosen because it is the “state of the art” but did they try other models (e.g. a GLM model) and see different results?

3. More information about the RGC types recorded is needed to evaluate the usefulness of the down-sampling encoding. Are the important features that need to be preserved during downsampling shared across RGC types? The relationship between image resolution and receptive field size should be made more clear early in the text, probably when discussing Figure 1. This would help the reader get a sense of how broadly this learned downsampling scheme could be used, in different cell types and in different species.

4. Related to (3) above, I found it a bit puzzling that there were so few recorded neurons per retina (on average, six per retina). Can the authors add some details about how units were identified and on what criteria units were excluded or included? What impact does this have on the learned networks that the recorded RGCs presumably did not tile visual space in many of the retinas being recorded? If some pixels are never seen by any RGC being recorded, how can the proper downsampling weights be learned for those pixels?

Minor points:

1. On page 2, end of first paragraph, text should be amended to clarify that RGCs project directly to other brain regions as well [e.g. SC], not just the LGN.

2. Make sure corrections for multiple comparisons are being used for significance tests when necessary (e.g. Fig. 1d)

Reviewer #2 (Remarks to the Author):

This is the review for "an actor-model framework for visual sensory encoding"
by Leong, Rahmani, Psaltis, Moser, and Ghezzi, 2023.

This paper describes a method to encode a retinal image so that it is down-sampled in an optimal manner (this manner is obtained by some learning algorithm) to fit the information to the limited number of electrodes in a retina prosthesis. The authors used a explanted retina from mouse for making and validating this method.

I find this topic interesting, however, the paper was written in a manner that is more like a presentation to authors' own colleagues who already know very well the overall project and the technical jargons. There is insufficient information for an outside reviewer to judge whether the content of this work, when written in a clearly understood way, is suitable for publication in Nature Communications. Reviewers of a Nature Communications paper should expect a paper written such that it is understandable by a Ph.D. in physical science (e.g., physics), perhaps the reviewer is in the general field of the authors, perhaps not, but the reviewer is most likely outside the lab of the authors or outside the special niche of this project.

I would be happy to review a substantially revised version of this paper.

To help the revision, I list some of my remarks and observations below:

(1) From the topic, at least a few equations are expected in the main text, after all, this is an optimization problem. However, there is not a single equation to describe what the problem is in a well-defined manner (rather than a hand-waving manner). In such a paper, equations should not be delegated to a few symbols or expressions in the figures or method section.

(2) Many of the technical terms in this paper are undefined. Please define them, and these could be connected with the mathematical formulation or description of the problem.

(3) It is stated on page 2 "Nevertheless, the literature on image encoding based on retinal modeling is scarce". This is unfortunate. There is a large body of literature, including textbook materials, on retinal image encoding, particularly in the area of understanding the neural transformation of the photoreceptor signals in the biological retina. The authors should get familiar with the literature and provide appropriate citations. A useful resource is the chapter "The efficient coding principle" in the textbook "Understanding vision: theory, models, and data" by Zhaoping published by Oxford University Press 2014.

(4) A mouse retina explant is used. This is not so suitable, since human vision is very different from mice vision. For example, human vision has a fovea where retinal sampling is much denser, but mouse does not have such a fovea. Human use vision as the dominant sense while mice' dominant sense is touch by whiskers and olfaction. If mouse rather than primate retina has to be used, please justify and explain how this impact the conclusion of the paper.

Reviewer #3 (Remarks to the Author):

The authors introduce a novel method for visual sensory encoding within the framework of an actor-model architecture, with a particular emphasis on image down-sampling. They demonstrate that employing the inherent computational capabilities of the retinal network for down-sampling results in superior performance compared to learning-free down-sampling techniques.

The study's results and methodology are indeed compelling, supported by a rigorous scientific approach. However, it is worth noting that the manuscript's structure may present challenges for readers. The research aim, objectives, and methodology are somewhat intertwined within the text, and the literature

review is relatively concise, and often embeds into the introduction. Additionally, the presentation of results could benefit from improved clarity and organization to enhance accessibility.

Reviewer #1 (Remarks to the Author):

In this manuscript, Leong and colleagues describe a method to learn a down-sampling encoding of natural images that preserves information important in retinal processing. They use a convolutional neural network model to predict RGC responses to flashed natural images, and then use this trained model to learn a second, downsampling network that minimizes changes in neural responses for a given downsampling factor. They then find that the learned downsampling network outperforms average downsampling in terms of neuronal reliability, and that what the learned downsampling seems to be doing is better preserving the contrast in the image compared to average downsampling.

The actor-model framework used here is very interesting and the biggest strength of the paper. The weaknesses of the paper include: a lack of clarity around what exactly the learned downsampling is doing, and whether it is actually better than a learning-free downsampling that preserves image contrast; and experimental concerns like lack of detail about cell types and small numbers of neurons per electrode recording. These weaknesses can be addressed with some in silico experiments, simple analyses, and small changes to data presentation and writing. Overall, this manuscript describes a very interesting framework for probing sensory encoding.

Major issues

1. My first set of suggestions has to do with the question of what the learned downsampling is doing to the images. The finding that the learned downsampling was effectively enhancing the contrast of the image relative to the average downsampling method was interesting, but it raises the question: is this all that the learned downsampling is doing, or is there more to it than that?

a. The authors should compare their learned downsampling to some unlearned downsampling methods that maintain the contrast of the original image. Using the model should be sufficient, and presenting new stimuli to ex vivo retinas is not necessary.

b. The authors should measure the effect on local contrast of the other unlearned downsampling methods in Fig. 4.

Authors' response

We thank the reviewer for pointing us into this direction. Among the learning-free methods originally proposed in Figure 4, the more advanced methods such as Lanczos filter preserve the contrast better compared to simple pixel averaging. Figure 7a in the revised manuscript (and reported here below) shows that some methods preserve contrasts even more than the actor-model framework. Our first conclusion was that contrast plays a crucial role in actor downsampled images. Therefore, this new result is counterintuitive since these learning-free methods did not perform as well as the actor network in terms of neuronal reliability (Figure 4 in the manuscript), despite having even higher local contrast preservation.

Figure 7. Effect of contrast on neuronal reliability. **a**, Each boxplot is the distribution of the pairwise difference in local contrast between actor downsampled images and downsampled images by a learning-free method. The first boxplot is the difference between actor downsampled and downsampled images by pixel averaging (as in Fig. 6b). **b**, Mean difference in neuronal reliability between actor downsampled images and downsampled images by pixel averaging with corrected contrast as a function of the contrast factor α . The red circle indicates the minimum of the curve for $\alpha = 1.5$. **c,d**, Distribution of the pairwise differences in local contrast (**c**) and neuronal reliability (**d**) between actor downsampled images and images downsampled by pixel averaging (left, as in Fig. 6b and Fig. 4) and images downsampled by pixel averaging with optimized contrast at $\alpha = 1.5$ (right). The box spans from the 25th to the 75th percentiles, the line is the median, and the whiskers are 1.5 times the interquartile range. p-values are reported for two-tailed tests as: n.s. $p > 0.05$, *** $p < 0.001$ and **** $p < 0.0001$.

Therefore, we wanted to better understand the role of contrast in effective image downsampling. For that, we assessed further the pixel averaging method by reducing/enhancing the contrast of the downsampled image before feeding it to the forward model for the evaluation of neuronal reliability. Results are reported here and in Figure 7b-d in the revised manuscript. In Figure 7b, $\alpha = 1$ indicates no changes to the contrast to the pixel average downsampled images while $\alpha < 1$ indicates a reduced contrast. $\alpha > 1$ represents an enhanced contrast. The gap in neuronal reliability between actor downsampled and average downsampled images widened when the contrast of average downsampled images is artificially reduced (Fig. 7b, contrast factor $\alpha < 1$). Conversely, when the contrast of average downsampled images is artificially increased ($\alpha > 1$), the gap in neuronal reliability decreases and minimizes for $\alpha = 1.5$. Moreover, when the contrast was increased further ($\alpha > 1.5$), the gap widened again. When the gap in neuronal reliability is minimized, the local contrast of the corrected average downsampled images is closer to the actor downsampled images although the difference is still statistically significant (Fig. 7c; $p < 0.0001$ two-tailed paired Wilcoxon test). This result indicates that there is an optimal level of contrast preservation which is reached by the actor network. Hence, other learning-free models fail to perform even though contrast was better preserved. Moreover, although increasing the contrast of the average downsampled images resulted in higher neuronal reliability, the actor network still performed better (Fig. 7d; $p = 0.0082$, two-tailed paired Wilcoxon test; $p = 0.0041$, one-tailed paired Wilcoxon test).

Hence, we hypothesize that contrast might not be the only factor learned by the actor network. To better understand the actor network, we plotted the weights of the kernels used in the actor network (Figure 8 in the revised manuscript and reported here below). From the hyperparameter tuning, the best result was obtained when training with 6 kernels, hence the 6 images correspond to the respective kernels. The six kernels learned by the actor network closely resemble a Mexican hat shape (Laplacian of a Gaussian) which is a common function in algorithms for edge detection. This function, or its fast approximation difference of Gaussians, is also traditionally used to model the center-surround organization of the RGC RF. This result is unexpected since the actor network was not specifically trained to mimic center-surround properties, yet it emerges when optimized based on the neuronal responses. The actor-model could integrate edge detection into the downsampling process by convoluting with a Mexican hat function. Hence, we hypothesize that this component also contributed to a more effective downsampling of the images in conjunction with optimizing the contrasts, leading to higher neuronal reliability.

Figure 8: CNN learned kernels. Visualization of the six kernels learned by the actor network. For each plot, the green profile corresponds to the line plot of the kernel across the horizontal midline (in green). The red profile corresponds to the line plot of the kernel across the vertical midline (in red). The kernels are categorized into two distinct groups: those with predominantly positive weights and those with predominantly negative weights, resembling center ON and center OFF properties.

c. The responses to average downsampled images (both in-silico and ex-vivo) seem to generally be weaker than responses to high-resolution or actor down-sampled images. This is consistent with the idea that average downsampling is just lowering the contrast, leading to weaker responses across all images. The authors should test whether this difference in overall response strength is present, and modify the

presentation of the data to make this more clear. For example, the far right scatter plots in 3a and 5a do not have X & Y axis labels that show which axis is which condition. I also suggest inserting a line of unity on these plots to show whether a condition drives higher or lower responses.

Authors' response

As requested, we modified the scatter plots in figs 3a and 5a.

About the weaker responses across the images, we performed a quantitative analysis on the mean firing of neurons across the images (Supplementary Fig. 2 in the manuscript and reported here). It is worth reporting that we found a statistically significant difference in the average response of RGCs to high-resolution, actor downsampled and average downsample images ($p < 0.0001$, Friedman test). In particular, the mean response to average downsample images is significantly lower than the average response to both high-resolution and actor downsample images ($p = 0.001$ for both comparisons, Nemenyi post-hoc test). On the contrary, the mean response of RGCs to actor downsample images was not significantly different from the response to high-resolution images ($p = 0.5398$, Nemenyi post-hoc test). This result confirms that actor downsampled images elicited neural responses more similar to high-resolution images.

Supplementary Figure 2. Response of RGCs in validation ex-vivo experiments. Quantification of the average response of RGCs upon presentation of high-resolution, average downsampled and actor downsample images. The response is the sum of the number of spikes occurring during image projection (400-ms window). The box spans from the 25th to the 75th percentiles, the line is the median, and the whiskers are 1.5 times the interquartile range. The black dots indicate outliers. p-values are reported as: n.s. $p > 0.05$ and ** $p < 0.01$.

d. If all that the actor down-sampling is doing is rescaling the contrast to match that of the original image, it leads one to wonder how this downsampling method would help in neural prosthesis design. For a prosthetic device, the conversion from contrast in the image to electrical stimulation is arbitrary anyway, so could you not achieve a similar improvement by just upping the gain from sensor to stimulator to achieve a few more spikes per unit contrast? The authors should give a description of either: 1. How the actor downsampling is doing something more interesting than a uniform rescaling of response gain or 2. How such a rescaling would assist in the design of a prosthetic device.

Authors' response

In principle, we agree with the reviewer that, after it has been found that contrast is an important factor, then a similar improvement might be obtained by playing with the stimulator chip. However, we are now able to investigate the possibility of tuning contrast for improving prosthesis design only because it was first revealed through training and analyzing the actor network, which narrowed down the directions that we can focus on.

Moreover, as discussed in point a, we believed that we have found that not only contrast but also the ricky wavelet used for convolution are important for the actor network. From these new results, we demonstrated that there is an optimal level of contrast enhancement. Therefore, the actor network is still useful in this case to find the optimal level of contrast, in conjunction with finding the optimal waveform to be used for the convolution. To search for this optimal waveform and contrast simply by tuning parameters of the stimulator manually will be too complicated in our opinion.

2. Is the learned downsampling dependent on the predictive power of the forward model? It is important to know whether we can expect improvements in forward modeling to yield changes in learned downsampling methods. The authors state that the CNN was chosen because it is the "state of the art" but did they try other models (e.g. a GLM model) and see different results?

Authors' response

Although GLM and LN models have been used traditionally to model the retina, McIntosh et al. [36] has shown that CNN performs significantly better than those models with white noise stimulus. For natural

images, the gap in performance is even larger [35]. Since we used natural images as the input, in our case, it is pertinent that the forward model chosen is suitable for modeling the retina presented with natural images. Hence we used a CNN. Extrapolating on this point, it implies that in the event that a better model of the retina is derived in the future, it will definitely boost the capability of the actor-model framework. Since the actor network is optimizing based on the output of the forward model, the closer the forward model is to the retina, the better it will be. Otherwise, the actor will be optimizing based on a more inaccurate model which will lead to a worse performance. This was also observed in Rahmani et al [48], where it has been shown that a powerful yet accurate model is pertinent.

3. More information about the RGC types recorded is needed to evaluate the usefulness of the down-sampling encoding. Are the important features that need to be preserved during downsampling shared across RGC types? The relationship between image resolution and receptive field size should be made more clear early in the text, probably when discussing Figure 1. This would help the reader get a sense of how broadly this learned downsampling scheme could be used, in different cell types and in different species.

Authors' response

One of the main considerations when designing the experiment was to generate a robust actor network. As such, this experiment was designed in a way invariant to the types of RGC recorded. A crucial property of CNNs is their weight sharing property. Weight sharing property refers to how each learned kernels are applied across the entire image. Therefore, it allows the kernels learned to be insensitive to the location in the image. Additionally, the feature learned will also be insensitive to the types of RGC types recorded since the kernels have to be optimized to contain features that are pertinent across the RGC types. This is particularly important because in a prosthesis device, it is not practical to identify which type of neuron is populated around the different electrodes and tune each electrode individually. By finding the pertinent features common across the RGC types, it simplifies the process. Following your comment, we have performed an analysis to prove the points mentioned above. Based on the STA analysis, we were able to obtain the type of RGC (on/off) and the size of the receptive field. We look at the neuronal reliability of actor downsampled images plotted against the type and RF size (Supplementary Fig. 1 of the revised manuscript and reported below). We did not find any statistical difference. This indicates that the actor model was not tuned specifically for any type or size of RGC and could generalize well.

This is particularly important because in a prosthesis device, it is not practical to identify which type of neuron is populated around the different electrodes and tune each electrode individually. By finding the pertinent features common across the RGC types, it simplifies the process. Following your comment, we have performed an analysis to prove the points mentioned above. Based on the STA analysis, we were able to obtain the type of RGC (on/off) and the size of the receptive field. We look at the neuronal reliability of actor downsampled images plotted against the type and RF size (Supplementary Fig. 1 of the revised manuscript and reported below). We did not find any statistical difference. This indicates that the actor model was not tuned specifically for any type or size of RGC and could generalize well.

Supplementary Figure 1. Neural reliability is neither dependent on the cell type nor the RF size. **a**, Difference in in-silico neuronal reliability of actor downsampled images to high-resolution images as a function of the cell type. **b**, Difference in in-silico neuronal reliability of actor downsampled images to high-resolution images as a function of the RF size. **c**, Difference in ex-vivo neuronal reliability of actor downsampled images to high-resolution images. **d**, Difference in ex-vivo neuronal reliability of actor downsampled images to high-resolution images as a function of the RF size.

4. Related to (3) above, I found it a bit puzzling that there were so few recorded neurons per retina (on average, six per retina). Can the authors add some details about how units were identified and on what criteria units were excluded or included? What impact does this have on the learned networks that the recorded RGCs presumably did not tile visual space in many of the retinas being recorded? If some pixels are never seen by any RGC being recorded, how can the proper downsampling weights be learned for those pixels?

Authors' response

We agree with the reviewer that the number of cells recorded per retina seems lower than the norm in literature. However, there are a few elements that must be considered.

For our methodology we had to include only cells that were stable over the entire recording period of about 5 hrs (to have sufficient data to train our neural network). This is now described in our methodological section. As a consequence, neurons that began being responsive after the stimulation program was started or stopped responding before the end of the stimulation program were discarded. This is now clarified as well.

A crucial difference between our recording approach and others is the electrode pitch of the MEA. In many papers, a smaller electrode pitch is used (our is 200 μm). We believe that higher electrode density allows SpyKING CIRCUS to better sort the spikes into different units.

The method section has been modified to provide more information about the spike sorting process.

In our opinion, the fact that the RGCs do not tile the visual space does not have a significant impact on the learned actor network. As mentioned before, the weight sharing property of CNN allows the actor network to learn the feature from the visual space that contains the RGCs, and subsequently apply the learned kernel throughout the whole visual space. This is also the reason why the results could generalize with the new set of retinas used in the validation ex-vivo experiments. Although we could not guarantee that the RGCs recorded in the new set of retinas would co-locate with the RGCs recorded in the first set of retinas, the fact that the actor network could generalize is testament to the robustness and invariant towards the types of RGCs and the location of the RGCs recorded.

Minor points:

1. On page 2, end of first paragraph, text should be amended to clarify that RGCs project directly to other brain regions as well [e.g. SC], not just the LGN.

Authors' response

Amended.

2. Make sure corrections for multiple comparisons are being used for significance tests when necessary (e.g. Fig. 1d)

Authors' response

We considered your suggestion. Just to clarify, figure 1d does not report a multiple comparison test. Statistical comparison is a paired test within each downsampling dimension. Comparisons between downsampling dimensions are not relevant here. Similar to the other figures.

Reviewer #2 (Remarks to the Author):

This is the review for "an actor-model framework for visual sensory encoding" by Leong, Rahmani, Psaltis, Moser, and Ghezzi, 2023.

This paper describes a method to encode a retinal image so that it is down-sampled in an optimal manner (this manner is obtained by some learning algorithm) to fit the information to the limited number of electrodes in a retina prosthesis. The authors used a explanted retina from mouse for making and validating this method.

I find this topic interesting, however, the paper was written in a manner that is more like a presentation to authors' own colleagues who already know very well the overall project and the technical jargons. There is insufficient information for an outside reviewer to judge whether the content of this work, when written in a clearly understood way, is suitable for publication in Nature Communications. Reviewers of a Nature Communications paper should expect a paper written such that it is understandable by a Ph.D. in physical science (e.g., physics), perhaps the reviewer is in the general field of the authors, perhaps not, but the reviewer is most likely outside the lab of the authors or outside the special niche of this project.

I would be happy to review a substantially revised version of this paper. To help the revision, I list some of my remarks and observations below:

(1) From the topic, at least a few equations are expected in the main text, after all, this is an optimization problem. However, there is not a single equation to describe what the problem is in a well-defined manner (rather than a hand-waving manner). In such a paper, equations should not be delegated to a few symbols or expressions in the figures or method section.

Author's response

Following this suggestion, we inverted figures 1 and 2. Also we started the results section with a formal introduction of the actor-network framework.

(2) Many of the technical terms in this paper are undefined. Please define them, and these could be connected with the mathematical formulation or description of the problem.

Author's response

We have worked on the terminology to make it more understandable to a general audience.

(3) It is stated on page 2 "Nevertheless, the literature on image encoding based on retinal modeling is scarce". This is unfortunate. There is a large body of literature, including textbook materials, on retinal image encoding, particularly in the area of understanding the neural transformation of the photoreceptor signals in the biological retina. The authors should get familiar with the literature and provide appropriate citations. A useful resource is the chapter "The efficient coding principle" in the textbook "Understanding vision: theory, models, and data" by Zhaoping published by Oxford University Press 2014.

Author's response

We believe here there is misunderstanding arising from the different use of technical terms, as remarked in point (2). We agree with the reviewer that there are many materials related to visual encoding within the retina itself. In such a sense, just a few sentences before it was stated: "Concurrently, there had been significant efforts to generate in-silico retina models that potentially could be used for efficient image encoding". This statement is reinforced in the discussion when we comment on the forward model.

However, in this article, image encoding (which is now named artificial image encoding), which we differentiate from retinal visual encoding (which is named visual encoding), refers to how encode the image into a prosthetic system which is different in nature and dimension compared to the natural system (the actor model). In this regard, our position remains unchanged that literature for image encoding which utilizes retinal modeling is scarce. We have modified the terminology to make it clear to the general audience and the sentence mentioned by the reviewer has been removed.

(4) A mouse retina explant is used. This is not so suitable, since human vision is very different from mice vision. For example, human vision has a fovea where retinal sampling is much denser, but mouse does not have such a fovea. Human use vision as the dominant sense while mice' dominant sense is touch by whiskers and olfaction. If mouse rather than primate retina has to be used, please justify and explain how this impact the conclusion of the paper.

Author's response

We agree with the reviewer that a NHP retina would be closer to humans. However, there are a number of reasons why we believe our approach is appropriate.

- It is not conceivable in our opinion to use NHP retinas for every research in vision, even if it will be a better model, in particular at an early stage. There are ethical concerns in the first place, but also availability of NHPs is currently very limited in Europe.
- Second and more importantly, from a scientific point of view, the interesting question is the capability of a CNN to generalize. We have shown in this paper that it can generalize within a sample (few neurons recorded) and between samples (different retinas) from the same species (mice). The next step for us is to determine if it can generalize as well across species, for example mice, rats, rabbits, NHPs etc. This would be a much more powerful result than just testing only in NHPs because they are closer to humans. However, this is also a new project that is beyond the scope of this paper.

This last comment was already mentioned in our outlook section. Now, it has been reinforced.

Reviewer #3 (Remarks to the Author):

The authors introduce a novel method for visual sensory encoding within the framework of an actor-model architecture, with a particular emphasis on image down-sampling. They demonstrate that employing the inherent computational capabilities of the retinal network for down-sampling results in superior performance compared to learning-free down-sampling techniques.

The study's results and methodology are indeed compelling, supported by a rigorous scientific approach. However, it is worth noting that the manuscript's structure may present challenges for readers. The research aim, objectives, and methodology are somewhat intertwined within the text, and the literature review is relatively concise, and often embeds into the introduction. Additionally, the presentation of results could benefit from improved clarity and organization to enhance accessibility.

Author's response

We have worked on the text to improve its clarity to a general audience.

One of the issues with nature's paper (and others) is that methods are appended at the end. In our view, it was necessary to anticipate some elements of methods (like figure 1a and text) to let the reader understand without going to the end of the paper and back. However, wherever possible, methodological points were removed from the text and moved to the method section.

REVIEWERS' COMMENTS

Reviewer #1 (Remarks to the Author):

The authors did a nice job of addressing my concerns. I commend them on the effort to really dig into the contrast question as it led to an interesting finding.

Reviewer #1 (Remarks on code availability):

The code repository is not accessible at this url. I suspect it is still set to private. I cannot evaluate the code.

Reviewer #3 (Remarks to the Author):

The revised version of the manuscript demonstrates a commendable level of enhancement. I am of the opinion that the research has reached a stage where it is well-suited for publication. It is noteworthy that although not all comments have been entirely addressed, the authors have undertaken a thorough justification of their viewpoints.

Reviewer #1 (Remarks on code availability):

The code repository is not accessible at this url. I suspect it is still set to private. I cannot evaluate the code.

Authors' response

We verified the link and it is working as a public repository.

Reviewer #3 (Remarks to the Author):

The revised version of the manuscript demonstrates a commendable level of enhancement. I am of the opinion that the research has reached a stage where it is well-suited for publication. It is noteworthy that although not all comments have been entirely addressed, the authors have undertaken a thorough justification of their viewpoints.

Authors' response

We thank the reviewer for the positive comment.